# Comparison of Pure and Crossbred Japanese Black Steers in Growth Performance and Metabolic Features from Birth to Slaughter at a Spanish Fattening Farm

**DOI:** 10.3390/ani12131671

**Published:** 2022-06-29

**Authors:** Juan M. Vázquez-Mosquera, Eduardo de Mercado, Aitor Fernández-Novo, Juan C. Gardón, José L. Pesántez-Pacheco, María Luz Pérez-Solana, Ángel Revilla-Ruiz, Daniel Martínez, Arantxa Villagrá, Francisco Sebastián, Sonia S. Pérez-Garnelo, Susana Astiz

**Affiliations:** 1Medicine and Surgery Department, Veterinary Faculty, Complutense University of Madrid, Puerta de Hierro Avenue s/n, 28040 Madrid, Spain; juvazq01@ucm.es (J.M.V.-M.); angelrevillaruiz@gmail.com (Á.R.-R.); 2Animal Reproduction Department, National Institute of Agronomic Research (INIA-CSIC), Puerta de Hierro Avenue s/n, 28040 Madrid, Spain; eduardo.mercado@inia.csic.es (E.d.M.); luz.perez@inia.csic.es (M.L.P.-S.); sgarnelo@inia.csic.es (S.S.P.-G.); 3Department of Veterinary Medicine, School of Biomedical and Health Sciences, Universidad Europea de Madrid, C/Tajo s/n, Villaviciosa de Odón, 28670 Madrid, Spain; aitor.fernandez@universidadeuropea.es; 4Department of Animal Medicine and Surgery, Veterinary and Experimental Sciences Faculty, Catholic University of Valencia-San Vicente Mártir, Guillem de Castro, 94, 46001 Valencia, Spain; jc.gardon@ucv.es; 5Faculty of Agricultural Sciences, School of Veterinary Medicine and Zootechnics, University of Cuenca, Av. Doce de Octubre, Cuenca 010220, Ecuador; jose.pesantez@ucuenca.edu.ec; 6Embriovet S.L, Polígono Industrial de Piadela II-8, Betanzos, 15300 A Coruña, Spain; daniel@embriovet.es; 7Institut Valencià d’Investigacions Agràries (IVIA), CV-315, Km 107, 46113 Valencia, Spain; villagra_ara@gva.es; 8Cowvet S.L, Avda. País Valenciano 6, 46117 Bétera-Valencia, Spain; cowvetsl@gmail.com

**Keywords:** metabolic markers, marbling, weight daily gain, Wagyu, Wangus

## Abstract

**Simple Summary:**

Cattle growth performance is a determinant of beef production. Nowadays, customers demand specialized, high-quality beef products produced according to stringent health and welfare standards. Intramuscular fat, or marbling, improves beef quality, and the Japanese Black (Wagyu) is the breed with the highest rates of marbling. Wagyu steers are reared under specific conditions in Japan, which may differ from the conditions in other countries, and these differences may affect animal well-being and, therefore, growth rates and beef quality. The current study shows that purebred Wagyu and crossbred Wagyu-by-Angus steers that were raised at a cow–calf operation and fattening system in Spain with no exercise restriction, high welfare, and a local diet high in olein content showed appropriate growth and fattening rates, health status, and metabolic development. Wagyu crossbred steers did not show substantially faster growth than purebred Wagyu animals, so they may not be as profitable as purebred Japanese Black in this type of production system.

**Abstract:**

Japanese Black (Wagyu) cattle produce high-quality beef. However, whether Wagyu steers can be profitably raised under conditions different than the traditional Japanese ones remains unclear. From 2018 to 2020, we raised 262 Wagyu purebred steers, 103 Wagyu-by-Angus (Wangus) crossbred steers, and 43 Angus-by-European (ACL) crossbred steers on a Spanish farm with high welfare standards and a locally sourced, high-olein diet. Factors and factors’ interactions impacting steer growth were analyzed using generalized linear models. ACL steers grew faster than the other two groups, with Wangus showing intermediate fattening and muscle development. Average daily weight gains (kg/day) were 0.916 for Wagyu, 1.046 for Wangus, and 1.293 for ACL during the weaning to growing period, and 0.628 for Wagyu, 0.64 for Wangus, and 0.802 for ACL during the growing to fattening phase. ACL showed the lowest marbling rates. Wagyu and Wangus usually showed higher cholesterol, triglycerides, and high-density lipoprotein than ACL. ACL calves may experience greater stress at weaning, as suggested by higher glucose, lactate, and β-hydroxybutyrate than the other groups. The results suggest that Wagyu and Wangus steers showed adequate growth, health, and metabolic development in this type of production system, with Wagyu purebreds probably being more profitable than Wangus crossbreeds.

## 1. Introduction

Beef quality attributes, such as color and fat content, depend on the characteristics of the breed and production system, and they are important for consumers [1]. Intramuscular fat, or marbling, improves beef quality in terms of juiciness, flavor, and tenderness, while modifying the color and the taste of the meat samples [2,3]. Certain breeds, such as Angus, show a high rate of marbling; therefore, their meat is considered to be of superior quality [4]. The bovine breed with the highest rates of marbling (around 30%) is the Japanese Black, also called Wagyu, which shows the most extensive infiltration of muscle by fat [5,6]. For the Angus breed, marbling ranges between 7% and 15%, while it reaches around 7% in Hereford [7], another breed of premium quality [7].

The Wagyu breed and its crosses are fed during long periods (28–30 months) [6] in order to achieve a high marbling rate. In addition, Wagyu animals require longer times than other breeds to grow and achieve adequate carcass size [8,9]. Beef producers may cross Wagyu with other breeds, such as Angus, to decrease production costs without notably reducing the final quality of the meat [9]. In fact, the commercial Wagyu-by-Angus crossbreed (“Wangus”) shows higher fattening performance and higher slaughter quality than purebred Angus or Hereford cattle [10,11]. 

In traditional Japanese Wagyu production systems, calves are generally raised until the age of 10 months in cow–calf operations, then moved to familiar feedlots, where they are fed indoors on a concentrate–based diet throughout the fattening period until slaughter. During this time, they live in small barns with restricted exercise and they receive nearly individual care [6]. These animals are known for their good temperament but also vulnerability to stress, especially as a result of environmental changes [12]. Free-moving animals that are kept in groups experience less stress than animals that are confined indoors to a restricted surface [13]; however, free-moving animals show lower fattening [13] and experience higher social stress [14], which may affect Wagyu production targets [15,16,17].

Wagyu cattle are produced not only in Japan, but also in other countries such as Australia and the USA. Wagyu meat production systems around the world generally try to imitate the Japanese conditions [18], but actual production conditions can vary, with potential consequences for animal well-being and beef quality. For example, environmental factors, such as pasture availability and quality, climate, food concentrate composition [19], steer-holding in large groups, and type of feeding according to growth stage [20,21,22], can influence feedlot performance and final meat characteristics [23].

Growth is a key factor for meat quality [6,23,24,25,26,27], in part because it affects the marbling rate [28]. Growth rate has additional economic importance in production systems because it influences the total amount of feed that is necessary and the length of the fattening phase [29].

Several studies have investigated the quality of meat and muscle physiology of Wagyu cattle in countries other than Japan [18,19,20,21,22], observing that modifications in the feeding systems can modify the growth rhythm and the meat quality of the animals. These findings highlight the need to clarify how Wagyu steers perform under different production conditions, taking into account their growth, welfare, and health. For example, recent studies have explored how to rear Wagyu steers via grazing, even in Japan, in order to reduce costs [30,31]. 

Blood metabolite concentrations are commonly used to assess the nutritional, welfare, and health status in cattle. Monitoring some of these parameters can help in assessing how efficiently the animals use the supplied nutrients [32]. In Wagyu cattle, the main metabolic parameters related to marbling and growth rates are cholesterol, glucose, and urea [33,34,35,36,37,38].

Despite the high economic value of Wagyu animals and their crosses [7], there is little information on the growth efficiency and performance of this breed, compared with the available information on crossbred animals, and local breeds managed under different conditions than in Japan. Therefore, the aim of the present study was to analyze the growth, performance, and metabolic parameters from birth to slaughter of fattening steers from three high-marbling bovine breeds: purebred Wagyu, crossbred Wagyu-by-Angus (Wangus), and crossbreeds of Angus with other European breeds, such as Charolais and Limousin (ACL). We examined these three animal types on a cow–calf operation and fattening system in Spain, where the animals had no exercise restriction, showed high welfare, and received a locally sourced, high-olein diet. We hypothesized that purebred and crossbred Wagyu steers could be healthy and profitably reared under these conditions outside of Japan, and that they would show different metabolic and growth traits than the European crossbreds under the same conditions.

## 2. Materials and Methods

### 2.1. Ethics Statement

This was an observational, prospective study of animals on one commercial farm (Fincas del Turia, Mudéjar–Wagyu farm, Teruel, Spain), and no experimental interventions were performed. Data were recorded during regular farming activities, without additional or invasive interventions. Therefore, no ethical approval was required as stipulated in the Spanish Policy for Animal Protection [39], which complies with the European Union Directive 2010/63/UE on the protection of research animals.

### 2.2. Animals and Farm

The farm is located in north-central Spain. Cows and cow–calf pairs were housed in a separate barn for the breeding herd, and calves stayed with their mothers (with 20 m^2^/cow of available shadowed surface) until weaning at 3 to 5 months of age. Male calves were routinely castrated by the veterinarian staff of the farm during the first two weeks following birth. Weaned calves were housed in growing barns (10 m^2^/animal) in groups of 14 to 18 animals/pen, up to the age of 22 months. During the last phase of fattening (>22 months of age until slaughter), steers were kept in groups of 10 animals/pen (20 m^2^/animal). All barns were open (natural ventilation), with anti-slip, concrete or soil floors and chopped straw bedding (Figure 1). 

The selection of the breed animals was based on that of the usual breeds observed in this kind of animal production system. The two breeds that are considered most profitable in terms of the growing rhythm and carcass performance are Charolais and Limousin. To increase the marbling rate among European breeds, crossing with Angus is the usual strategy. Therefore, the following breed groups were considered: purebred Wagyu steers, Wagyu-by-Angus crossbred animals (Wangus), and crossbreeds of Angus with Charolais and Limousin (ACL).

The weight, height, metabolic parameters, growth patterns, and fatness of 408 steers (262 Wagyu, 103 Wangus, 43 ACL steers) were studied from 2018 to 2020. Data for all available animals at the farm were collected at four time points: (1) after weaning (WEAN), when the animals were around 4.5 months old (192 Wagyu, 56 Wangus, 19 ACL); (2) at the end of the growth period (GR), when the animals were around 13.5 months old (178 Wagyu, 85 Wangus, 38 ACL); (3) at the end of the finishing period (FIN) for ACL animals, when the animals were 22–26 months old, because the ACL animals had to be slaughtered earlier than the Wagyu animals (60 Wagyu, 55 Wangus, 32 ACL); and (4) at the end of the fattening period or the “slaughter phase” (SL) for the Wagyu and Wangus animals, when they were around 32 months old (11 Wagyu, 28 Wangus).

As part of this longitudinal analysis, we defined two subsets of animals in order to maximize the number of steers analyzed during the various time points. We chose a subset of 139 Wagyu, 50 Wangus, and 17 ACL animals that we followed from weaning until the growth phase, and a second subset of 57 Wagyu, 49 Wangus, and 31 ACL animals that we followed from the growth phase to the end of the finishing period. A total of 25 Wagyu, 14 Wangus and 14 ACL steers were included in both subsets. Therefore, the study had certain limitations for the longitudinal design, such as the number of animals per group, the wide range of the time interval for measuring the body fattening rate, and the limited number of animals that were evaluated during the slaughtering phase for the ACL group. However, the value of the data is enhanced by having been recovered under real commercial conditions.

The nutritional management of the animals included ad libitum water and diets adjusted to their requirements [40], depending on the fattening phase. Up to weaning, the calves had ad libitum access to starter feed (Table 1); from weaning onwards the calves received a total mixed ration (TMR) to avoid the selection of raw materials; From weaning to 10 months of age they received a dry TMR feed, which we call “growth dry-TMR”; from 10 to 22 months of age, the diet was wet TMR (“fattening wet-TMR”); and from 22 months of age to slaughter the diet was a dry TMR feed (“finishing dry-TMR”). The specific feed compositions are detailed in Table 1 and Table 2. 

### 2.3. Evaluation of Weight, Growth Pattern, and Fatness

At the four time points of the study, the animals were weighed using an electronic balance with an accuracy of 0.5 kg (Tru-Test, Auckland, New Zealand), while their height from the floor to the tail gate was determined using a measuring tape.

The average daily weight gain was calculated for the periods from WEAN to GR (weaning to growth interval), from GR to FIN (growth to finishing interval), and from WEAN to FIN phase (weaning to finishing interval). 

The body fattening rate was measured based on monthly ultrasound exams in a representative sample of the animals, using the PieQuip technology and a MyLab One^®^ ultrasound system (Esaote, Barcelona, Spain) with an animal science probe (ASP) of 18 cm length and a frequency of 3.5 Mhz [41,42]. Measurements were assigned to four time intervals (8–14, 15–16, 17–18, and 19–20 months of age), which were defined to minimize the variations within each interval. In animals for which multiple measurements were made during the same interval, the mean of those measurements was used for that interval. 

The following ultrasound features were analyzed because of their association with growth and fat deposition [42]: depth of the gluteus medius muscle (GMD, in mm), measured at the “P8” site, on the rump, at the intersection of a line going forward from the pin and a line down from the high point in the hindquarter, at the intersection of the gluteus medius and biceps femoris muscles; ribeye area (REA, in cm^2^) of the longissimus dorsi muscle, measured between the 12th and 13th ribs; back fat (BF, mm), defined as the thickness of fat between ribs 12 and 13; rump fat thickness (RF, mm), also measured at the P8 rump site; and percentage of intramuscular fat (IMF, corresponding to marbling), which was calculated by the ultrasound software when the transducer was situated at longissimus dorsi between ribs 12 and 13.

No ultrasound data were available to calculate REA or IMF of animals older than 22 months, because the layer of dorsal fat was thicker than 20 mm, which prevented high-quality ultrasound imaging.

### 2.4. Evaluation of Metabolic Status

Metabolic status was assessed based on plasma analysis from blood routinely sampled as part of health assessments on the farm during the growth phase. Samples were obtained by coccygeal vein puncture and collected into standard 10-mL EDTA vacuum tubes (Vacutainer^®^ System Europe; Becton Dickinson, Meylan, France). Blood samples were centrifuged at 4500× *g* for 15 min, and the plasma was stored in polypropylene vials at −80 °C until analysis. Plasma was evaluated with a clinical biochemistry analyzer (Konelab 20; Thermo Fisher Scientific, Waltham, MA, USA) according to the manufacturer’s instructions. Levels of total cholesterol (TC), triglycerides (TG), high-density lipoprotein (HDL), low-density lipoprotein (LDL), glucose (GLU), fructosamine (FRU), lactate (LAC), β-hydroxybutyrate (BHB), non-esterified fatty acids (NEFAs), and urea were measured. 

### 2.5. Statistical Analyses

Data were analyzed using SPSS^®^ 25 (IBM, Armonk, NY, USA). Changes over time were assessed using analysis of variance for repeated measures. Generalized linear models (GLMs) with Greenhouse–Geisser correction were used to assess interactions between the studied factors; in these models, breed and time were considered fixed factors. The normality of variables was assessed using Kolmogorov–Smirnov and Shapiro–Wilk tests. Variables showing a skewed distribution were reported as median and range, and intergroup differences in these variables were assessed for significance using the Kruskal–Wallis or Mann–Whitney U tests for independent samples. Potential pairwise relationships in skewed variables were assessed using the Spearman test. Continuous variables showing a normal distribution were reported as mean ± standard deviation. Differences associated with *p*-values ≤ 0.05 were considered significant.

## 3. Results

The number of animals assessed, with their corresponding ages, are summarized in Table 3 and Table 4.

### 3.1. Growth Pattern (Weight, Height, and Average Daily Gain)

The weight and height of the studied animals at the different time points are depicted in Figure 2 and Figure 3.

The weights and heights were higher for ACL animals than for the other groups, especially at WEAN and GR. However, the values were similar among the three groups at FIN. At SL, the Wagyu and Wangus animals showed similar weights and heights.

Changes in weight and height over time are shown in Figure 3. The breed, the time, and the interaction between breed and time had a clear effect on these variables, with ACL animals growing faster than the other two groups from weaning to growing, although ACL growth slowed from the age of 13.5 months (Figure 3B,D).

The average daily weight gains are summarized in Table 5.

Consistently with the observations shown in Figure 3, these results revealed a pattern of higher weight gain in ACL animals, which grew fastest during the interval from WEAN to GR. In the interval from GR to FIN, the growth rate and the final weight at FIN were similar among the breeds.

### 3.2. Ultrasound Evaluation of Fat and Muscle Deposition

Ultrasound measurements of the parameters related to fat deposition and muscle growth are summarized in Figure 4.

Deposition of fat and muscle followed different trajectories in the WY and ACL groups, while the WN group showed an intermediate trajectory. ACL steers presented the highest values of GMD (Figure 4A) in all intervals. The interaction between breed and time significantly affected growth (*p* < 0.001), with the ACL animals growing differently than the WY and WN steers over time. The WY animals showed the lowest muscle development based on REA (*p* < 0.001, Figure 4B).

Fat deposition and muscle development varied significantly among the breeds. The WY group presented the lowest BF values (Figure 4C), while the ACL animals showed the highest fat accumulation at the P8 RF (Figure 4D), especially late in the fattening phase. The IMF (Figure 4E) was affected by breed, with the lowest values observed in the ACL steers. Images on the final marbling stage of the beef are shown in Figure 5.

### 3.3. Metabolic Status

The assessment of the metabolic parameters is summarized in Table 6, and their trajectories over time are depicted in Figure 6 and Figure 7.

The ACL steers showed higher values of energy-related metabolites than the other two groups, while the values of lipid-related metabolites were lower in the ACL steers, especially earlier in life. The WY and WN animals showed similar levels of plasma BHB, GLU, and LAC, and similar levels of TC in WEAN and FIN. However, the WN animals had higher values of TG, LDL, and HDL, and lower values of NEFA and FRU than their WY counterparts in the late fattening phase. Similarly, the WN animals showed higher levels of TC than the WY animals at GR. The WN animals showed the highest levels of urea throughout the study. Before slaughter, the WY and WN animals differed significantly in LDL and urea content.

In the ACL animals, the BHB, GLU, and LAC levels decreased from weaning to growth (Figure 6A,E,G). GLU decreased less sharply in the WY and WN steers, while BHB, NEFA, and LAC remained stable or even increased with time. FRU decreased in all animals, but less sharply in the ACL steers. In the second interval studied, from GR to FIN, BHB and LAC continued to slightly decrease in the ACL animals (Figure 6B,H), with a less marked slope, while NEFA increased in the ACL and WY steers. NEFA, GLU, and FRU decreased in the WN steers (Figure 6D,F,J), while BHB and LAC increased or remained stable, respectively (Figure 6B,H). In the WY steers, NEFA and FRU values increased, while GLU and BHB decreased with time. Other metabolites remained stable.

From WEAN to GR, LDL decreased in all three groups (Figure 7E), while TC and HDL decreased in the WY and WN animals but increased in the ACL steers (Figure 7A,C). However, from GR to FIN, TC, HDL, and TG increased in the WY and WN animals (Figure 7A,D,H). Urea did not change significantly over time in any breed from WEAN to GR, but it changed significantly from GR to FIN, with a slight increase only in the ACL steers.

## 4. Discussion

Despite the high economic value of Wagyu animals and their crosses, little is known about their growth efficiency and performance under conditions different than those in Japan. This study aimed to describe the growth, performance, and metabolic parameters from birth to slaughter of steers from three groups of high-marbling bovine breeds on a Spanish cow–calf operation and fattening system. As we hypothesized, pure and crossbred Wagyu steers were healthy and profitably reared under Spanish conditions, and they showed different metabolic and growth traits than those of European crossbred animals in the same production system.

Our results confirm that the ACL steers grew faster than the Wagyu cattle and its crossbreeds from weaning onwards, thereby achieving an earlier slaughter weight. Indeed, Limousine and Charolais breeds grow faster than other European breeds [43], due to their faster nutrient conversion, regardless of the fattening period [8,9]. We found that the average daily gain was consistently around 29% higher for the ACL animals than for Japanese animals over all the intervals that were examined. Similar results were previously reported in comparisons of several European breeds with Wagyu [44]. In our study, Wangyu cattle presented the lowest daily weight gain, as previously described [6]. Therefore, our results demonstrate that Wagyu animals raised in a Spanish production system can show comparable performance as the same breed raised under traditional Japanese conditions.

In vivo evaluated intramuscular fat, the most important characteristic for the high value of Wagyu beef, was similar in the Wangus and Wagyu animals, and higher than that of the ACL steers. The IMF assures the high price and extra quality of this kind of beef [45], compensating for the higher cost of the long fattening period [45]. Further studies should examine how the differences that we found correlate with the market value of beef from different animal breeds. 

We confirmed the adequate health status of all animals on the Spanish farm at all time points, with all metabolites falling within physiological ranges [37,38,39,40,41,42,43,44,45,46], although one report indicated that Wagyu cattle may reduce feed intake during summer due to heat stress [47]. Therefore, this type of production system, under Spanish climatic conditions with no restriction of exercise and with a locally sourced, high-olein diet, is appropriate for rearing and fattening Wagyu and Wangus animals.

Ultrasound scanning is helpful in determining whether there is a localized or infiltrated marbling rate in beef cattle and in adjusting the optimal slaughter time [48,49,50]. In our ultrasound studies, the ACL animals showed greater muscle development (GMD and REA) and subcutaneous fat deposition (BF and RF) than the Wagyu steers, until 20 months of age. These values are comparable with those of a previous study of ACL animals in Spain [51]. The slight differences between that previous study and ours can be explained by the fact that our ACL animals were crossbred with Angus, which resulted in enhanced fattening. Our ultrasound results are also similar to those from a previous study of Angus in the USA [52], although the level of fattening was slightly higher in our animals. This similarity is not surprising, given the high level of olive oil common to the diets. This higher oil content in the diets led to an earlier slaughter date for the ACL animals, in order to ensure carcass quality [53,54,55].

Ultrasound studies on Japanese breeds are limited, due to difficulties in obtaining measurements shortly before slaughter [48,52,56]. Indeed, we could not obtain ultrasound information of sufficient quality from Wagyu or Wangus animals that were older than 22 months. Therefore, we could not estimate the intramuscular fat of older animals or use such an estimate to optimize the slaughter time. For such situations, it may be possible to use appropriate models. For example, one decision model [57] aimed to predict subcutaneous fat depths in lean bovine breeds, while another model [58] aimed to estimate changes in marbling in Wagyu steers based on longitudinal ultrasound measurements. The model proposed by Walmsley et al. [57] was not adapted for Black Japanese cattle; however, the very recent model described by Tokunaga et al. [58] showed a negative relationship between the marbling score of carcass meat and the maturity level of beef of animals 24 months old, indicating that selection for an improved carcass beef marbling score would produce responses in an unfavorable direction for the beef maturation. Thus, further studies on Black Japanese would enhance clarity in this area.

We confirmed that the Wagyu and Wangus steers required longer fattening periods than that of the ACL animals [8,9] before they can be slaughtered. In fact, the Wangus animals showed a growth pattern quite similar to that of the purebred Wagyu animals, whereas we would have predicted that they would show a pattern between that of the Wagyu and ACL animals. Korean cattle closely related to Wagyu [59] (i.e., they show high heritability of bovine growth (0.58–0.76) [60]), and the heritability of body weight is higher in the Wagyu animals than in other breeds, e.g., 0.32–0.52 for Angus cattle [61], 0.19–0.25 for Simmental cattle [62], and 0.26–0.35 for the US Meat Animal Research Center III composite breed [63]. This higher heritability could explain the observed similarity between the Wangus and Wagyu animals’ growth patterns in our study. Our results imply that Wangus animals may not be as profitable as Wagyu steers, because the Wangus and Wagyu animals did not differ in terms of the length of the required fattening period. Even if the beef quality is similar, crossbred products do not command the same prices as purebred ones [45]. 

Ultrasound analysis identified two aspects in which the characteristics of Wangus steers may be between those of the Wagyu and ACL animals: (1) from 15 months of age, the Wangus animals developed GMD in a manner similar to that of the Wagyu animals, but their RFA resembled that of the ACL animals, and (2) the BF of the Wangus animals was comparable to that of the ACL steers, whereas the RF of the Wangus animals resembled that of Wagyu steers (Figure 4). These results may reflect the better carcass conformation of the Wangus animals than that of the Wagyu counterparts, which indicates that the Wangus steers are a more efficient breed in terms of carcass development [64]. 

Our results show clear metabolic differences across the three animal groups, primarily between the ACL animals and the other two groups. During the growth phase, the ACL animals showed greater changes (i.e., decreases) in glucose and lactate with time, while the Wagyu and Wangus animals seemed to maintain more stable levels. The ACL steers showed higher values of glucose at weaning and at 22 months than the other two breeds. A previous work [65] found that Wagyu steers have higher plasma insulin and lower plasma glucose than Holstein steers during the fattening period, a finding that was similar to our results. Moreover, our results concur with a previous study [66] that reported that plasma insulin concentration correlates positively with carcass fat and negatively with carcass muscle.

Lactate levels decreased more sharply in the ACL steers than in their Japanese counterparts. Because sudden increases in lactate can be considered a marker of acute stress [67], our results imply that the animals in our study experienced adequate welfare and health status. The higher levels of lactate at weaning in the ACL animals may reflect a higher level of stress during this phase, perhaps reflecting the quieter temperament of the Japanese breeds, especially those animals whose muscles have been more infiltrated by fat [68].

The high rate of marbling in the Japanese breeds is related to certain blood metabolites [33,34,35,36,37,38]. Plasma TC may be higher in the Wagyu animals, compared with Holstein steers [65], and our TC and TG findings are consistent with this idea. Interestingly, the Japanese steers in our study showed higher levels of HDL and lower levels of LDL than those of the ACL steers, indicating a lower atherogenic lipidic profile [69,70]. This may mean that the Japanese breeds have a metabolism that is better adapted to their high body fat content.

The BHB level was significantly higher in the ACL animals than in the two other breeds at weaning, perhaps reflecting a higher level of stress in the ACL animals, which can reduce their feed intake [71]. The BHB level in Japanese animals was slightly higher at other time points than at weaning, but it was always within the physiological range [37,38,39,40,41,42,43,44,45,46]. We found no significant differences in NEFA levels between the ACL and Wagyu animals, which was similar to the finding of another study that compared Holstein and Wagyu [65]. The similar NEFA levels across our three animal groups, notwithstanding the high content of vegetable oil in the diet, supports the idea that nutritional management on the Spanish farm was adequate. 

The Wagyu and Wangus steers in our study—particularly the Wangus animals—showed higher levels of urea than the ACL animals, and the levels exceeded the physiological limit proposed by Oyomaru et al. (2016) [37]. Our results suggest that Wangus animals engage in greater protein metabolism than Wagyu animals, which would lead to greater muscle mass, which we observed by ultrasound. This possibility should be explored in future work. 

## 5. Conclusions

A Spanish cow–calf operation and fattening system involving no exercise restriction, high animal welfare, and a locally sourced, olein-rich diet can support satisfactory growth rates, health, and metabolic development of purebred Wagyu and crossbred Wagyu-by-Angus steers. The performance of these two breeds differs from that of European high-marbling crossbreeds. Wangus animals do not grow substantially faster than Wagyu in this type of production system, so they may not be as profitable under these conditions.

## Figures and Tables

**Figure 1 animals-12-01671-f001:**
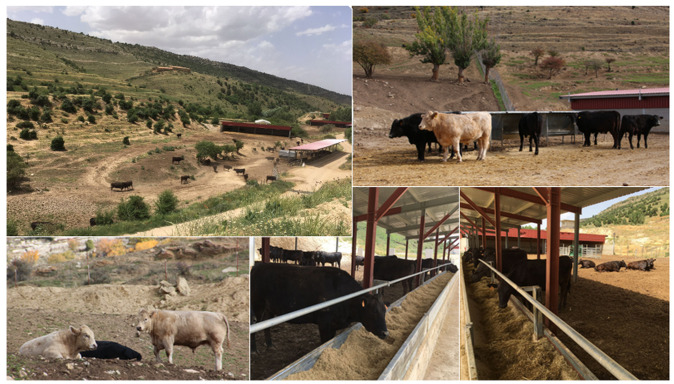
Fincas del Turia, Mudéjar–Wagyu farm. Purebred Wagyu steers, Wagyu-by-Angus crossbred animals (Wangus), and crossbreeds of Angus with Charolais and Limousin (ACL) are masted in open barns.

**Figure 2 animals-12-01671-f002:**
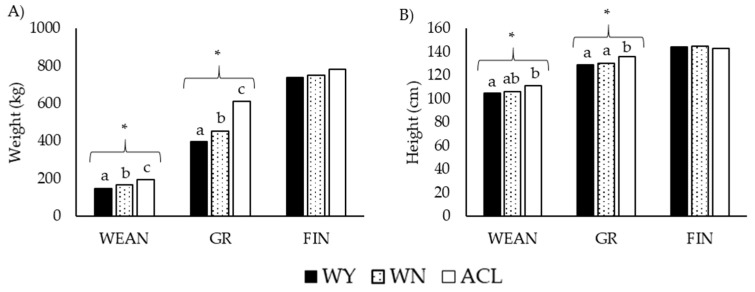
Transversal approach of (**A**) weight and (**B**) height of steers at three time points during fattening. Black bars: purebred Wagyu animals (WY); dotted bars: Wangus animals (Wagyu-by-Angus; WN); white bars: Angus-by-Charolais or Angus-by-Limousin crossbred animals (ACL). Abbreviations: WEAN: after weaning; GR: growing period; FIN: finishing period. Parameters are expressed as median values. Different superscripts a, b, and c indicate significant differences with * *p* < 0.001 by Kruskal–Wallis test.

**Figure 3 animals-12-01671-f003:**
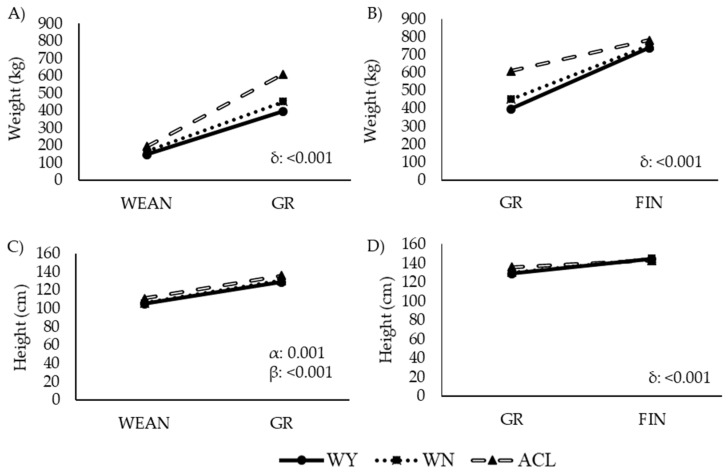
Changes in (**A**,**B**) weight and (**C**,**D**) height of steers during two intervals within the fattening period. Abbreviations: ACL: crossbred animals Angus-by-Charolais or Angus-by-Limousin; WEAN to GR: weaning to growing interval; GR to FIN: growing to finishing interval; WN: Wangus animals (Wagyu-by-Angus) breed; WY: fullblood Wagyu animals. α indicates significant effect by breed; β indicates significant effect by time; δ indicates significant effect by the interaction between breed and time. Parameters are expressed as median values.

**Figure 4 animals-12-01671-f004:**
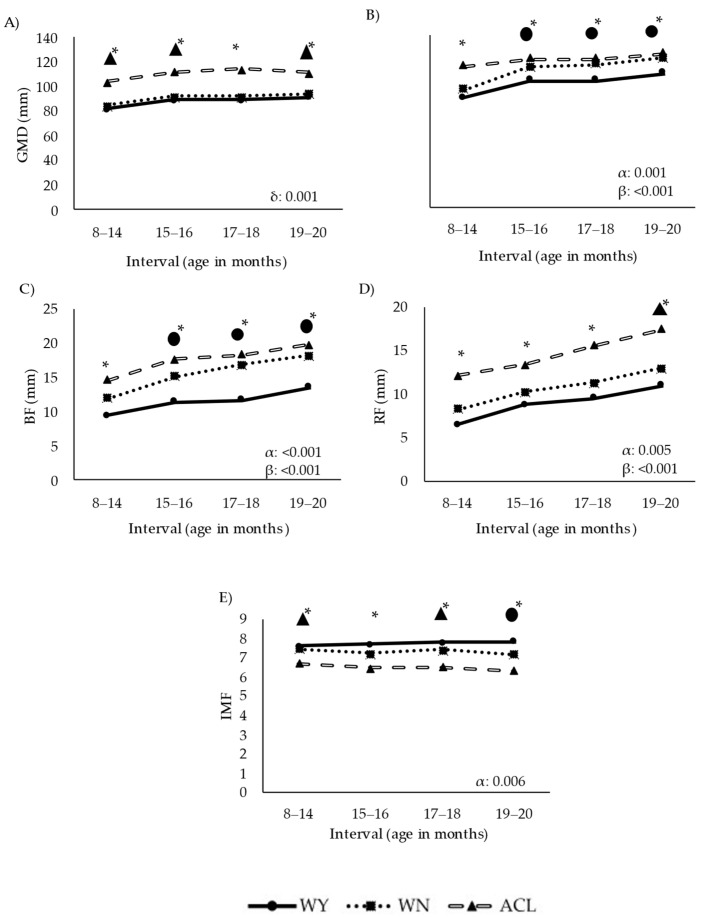
Longitudinal approach of ultrasound estimates of fat and muscle deposition over time in steers during the fattening period. Abbreviations: Panel (**A**): GMD, depth of the gluteus medius muscle measured at point “P8” rump site; Panel (**B**): REA, ribeye area of the longissimus dorsi measured between the 12–13th ribs; Panel (**C**): BF, back fat thickness measured between the 12–13th ribs; Panel (**D**) RF, rump fat thickness at the “P8” rump site; Panel (**E**): IMF, intramuscular fat estimation (numerical values provided by the software); ACL, crossbred Angus-by-Charolais or Angus-by-Limousin; WN, Wangus animals (Wagyu-by-Angus) breed; WY, fullblood Wagyu animals. One asterisk indicates that all breeds differed significantly from one another (*p* < 0.001). ●* indicates that the breed WY differed significantly from the other two (*p* < 0.001). ▲* indicates that the breed ACL differed significantly from the other two (*p* < 0.001). α indicates significant effect of breed; β indicates significant effect of time; δ indicates significant effect of the interaction between breed and time. Parameters are expressed as median values.

**Figure 5 animals-12-01671-f005:**
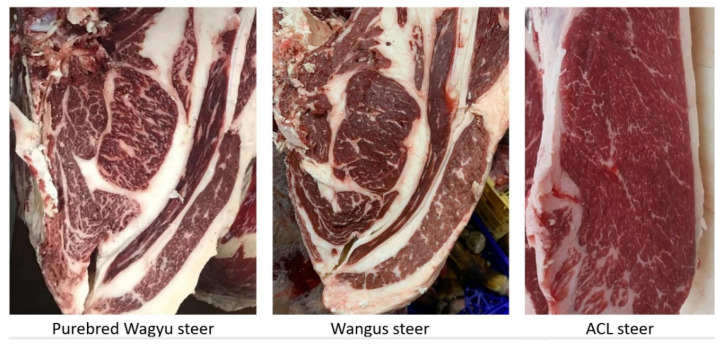
Marbling of beef of the different breed groups: pure bred Wagyu steers, Wagyu by Angus crossbred animals (Wangus), and crossbreeds of Angus with Charolais and Limousin (ACL).

**Figure 6 animals-12-01671-f006:**
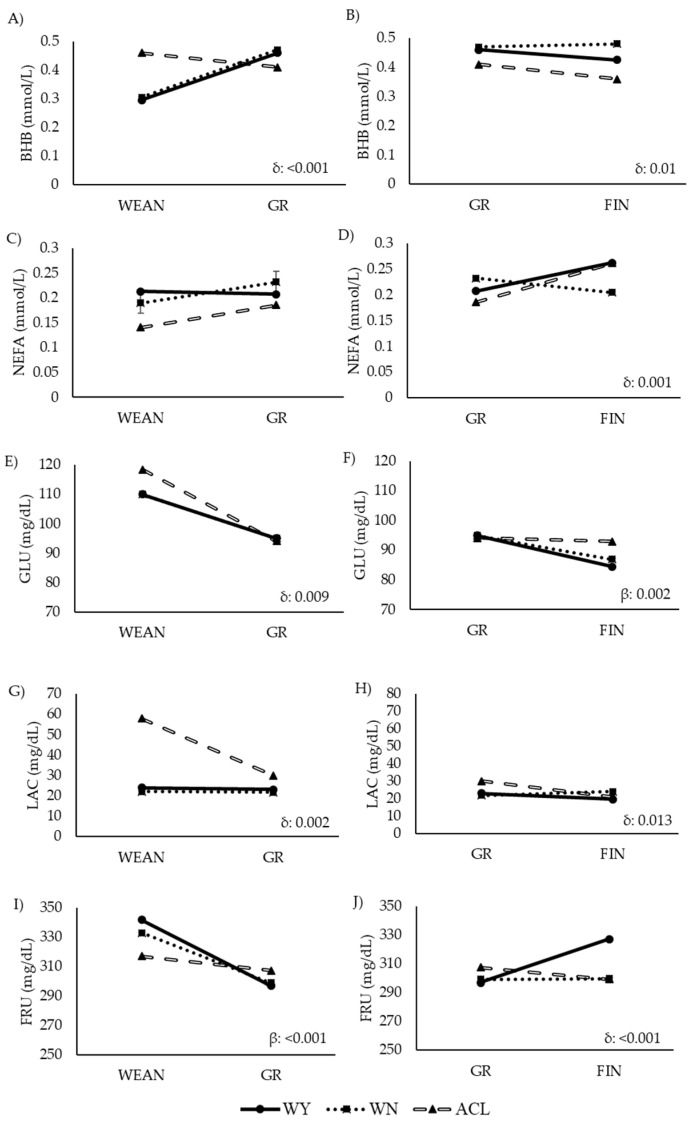
Comparison of levels of energy-related metabolites in steers at two time points during the fattening period (longitudinal approach). Abbreviations: ACL: crossbred animals Angus-by-Charolais or Angus-by-Limousin; BHB: β-hydroxybutyrate: Panels (**A**,**B**); FIN: finishing period; FRU: fructosamine: Panesl (**I**,**J**); GLU: glucose: Panels (**E**,**F**); GR: growing period; LAC: lactate: Panels (**G**,**H**); NEFA: non-esterified fatty acid: Panels (**C**,**D**); WEAN: after weaning; WN: Wangus animals (Wagyu-by-Angus); WY: fullblood Wagyu animals. Parameters are expressed with median values. β indicates significant effect of time; δ indicates significant effect of the interaction between breed and time.

**Figure 7 animals-12-01671-f007:**
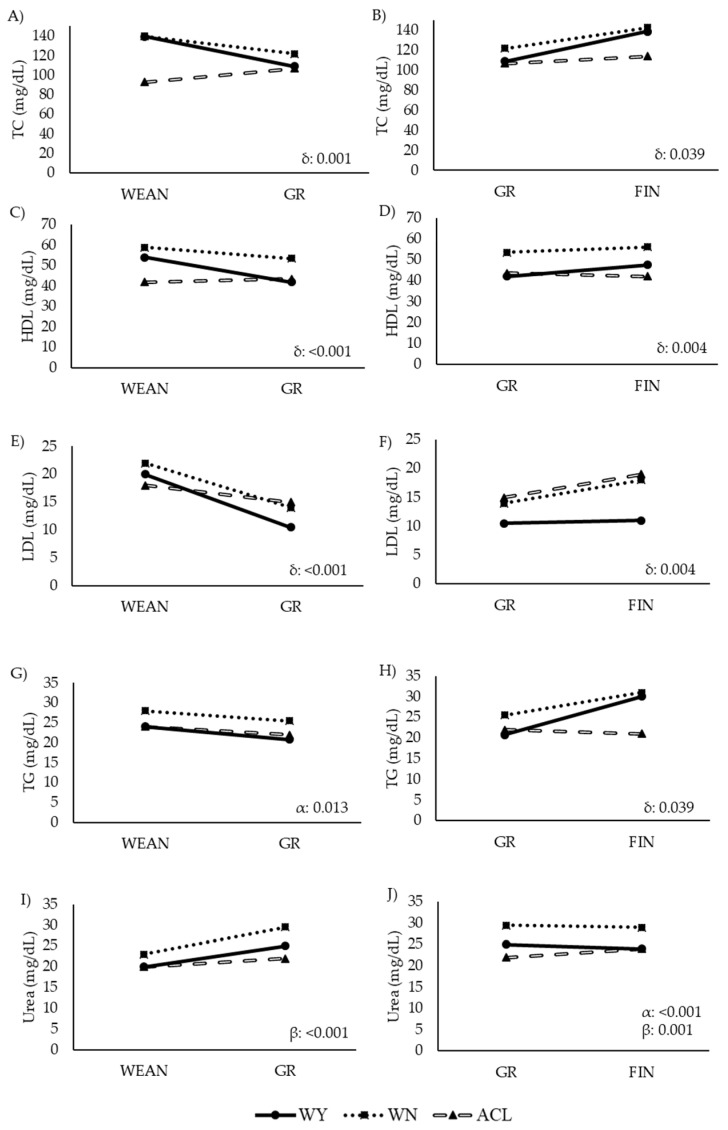
Comparison of levels of lipid-related metabolites and urea in steers at two time points during the fattening period (longitudinal approach). Abbreviations: ACL: crossbred animals Angus-by-Charolais or Angus-by-Limousin; GR to FIN: growing to finishing interval; HDL: high density lipoprotein cholesterol: Panels (**C**,**D**); l; LDL: low density lipoprotein cholesterol: Panels (**E**,**F**); TC: Serum total cholesterol: Panels (**A**,**B**); TG (mg/dL): triglycerides: Panels (**G**,**H**); Urea (mg/dL): Panels (**I**,**J**); WEAN to GR: weaning to growing interval; WN: Wangus animals (Wagyu-by-Angus); WY: fullblood Wagyu. Parameters are expressed with median values. α indicates significant effect of breed; β indicates significant effect of time; δ indicates significant effect of the interaction between breed and time.

**Table 1 animals-12-01671-t001:** Diet composition (percentage of dry matter) provided to all steers in the study during the weaning period.

Nutrient	Starter Feed
Crude fiber (%)	7.1
Ash (%)	4.9
Crude protein (%)	17.2
Crude fat (%)	4
Ca (%)	0.71
P (%)	0.34
Na (%)	0.14
Vitamin E (UI/kg)	26
Vitamin A (KUI/kg)	16
Vitamin D (KUI/kg)	2.2
Co (mg/kg)	0.2
Mn (mg/kg)	100
Zn (mg/kg)	90
Se (mg/kg)	0.46
I (mg/kg)	1
Cu (mg/kg)	26

Abbreviations: Ca: calcium; Co: cobalt II carbonate; Cu: copper II sulfate pentahydrate; I: potassium iodide; Mn: manganese oxide; Zn: Zinc, Na: sodium; P: phosphorus; Se: sodium selenite; TMR: total mixed ration.

**Table 2 animals-12-01671-t002:** Diet composition (percentage of dry matter) given to all steers in the study during the growth and fattening periods.

Nutrient	Growth Dry-TMR	Fattening Wet-TMR	Finishing Dry-TMR
Humidity (%)	10.90	67.79	11.35
Dry matter (%)	89.09	32.21	88.65
Crude fiber (%)	17.49	22.89	11.15
Ash (%)	6.64	6.95	5.58
NDF (%)	35.62	51.09	26.43
Crude protein (%)	16.34	13.52	13.19
Crude fat (%)	3.50	3.94	6.98
ADF (%)	22.73	30.12	14.16
Ca (%)	0.852	0.717	0.628
P (%)	0.286	0.286	0.292
Na (%)	0.175	0.304	0.219
Cl (%)	0.323	0.479	0.419
Mg (%)	0.34	0.331	0.219
K (%)	1.045	1.14	0.778
S (%)	0.183	0.25	0.172
Vitamin E (mg/kg)	32.32	14.693	41.62
Vitamin A (KUI/kg)	2.508	1.189	4.23
Vitamin D (KUI/kg)	0.702	0.333	1.18
NFC	37.898	24.50	47.83
Starch (%)	27.83	7.51	40.46
Fodder (%)	17.78		
ME (kcal/kg)	2785.93	3651.41	3026.4
TDN (%)	73.58	62.66	80.93

Abbreviations: ADF: acid detergent fiber; Ca: calcium; Cl: chlorine; K: potassium; ME: metabolizable energy; Mg: magnesium; Na: sodium; NDF: neutral detergent fiber; P: phosphorus; S: sulfur; NFC: non-fibrous carbohydrates,calculated according to Institut National de la Recherche Agronomique (INRA) guidelines); TDN: total digestive nutrients; TMR: total mixed ration.

**Table 3 animals-12-01671-t003:** Number and ages of fattening steers at each of the study’s time points (transversal approach).

	Wagyu		Wangus		ACL	
Time Point	n	Age (Months)	n	Age (Months)	n	Age (Months)
WEAN	192	4.2 ± 0.95	56	4.5 ± 0.83	19	4.4 ± 0.92
GR	178	12.8 ± 1.75	85	14.1 ± 2.52	38	14.3 ± 1.59
FIN	60	28.6 ± 3.65	55	27.9 ± 3.72	32	20.1 ± 2.53
SL	11	34.9 ± 4.46	28	32.8 ± 6.46		

Abbreviations: ACL: Angus-by-Charolais or Angus-by-Limousin crossbred animals; FIN: finishing period; GR: growing period; SL: slaughtering phase; WEAN: after weaning. Ages are expressed as mean ± standard deviation.

**Table 4 animals-12-01671-t004:** Number and ages of fattening steers during each growth interval and time point (longitudinal approach).

		Wagyu		Wangus		ACL	
Interval	Time Point	n	Age (Months)	n	Age (Months)	n	Age (Months)
WEAN to GR	WEAN	139	4.6 ± 0.77	50	4.7 ± 0.75	17	4.6 ± 0.93
	GR		12.5 ± 1.74		12.7 ± 1.49		13.7 ± 0.99
GR to FIN	GR	57	13.8 ± 0.84	49	15.3 ± 2.36	31	14.3 ± 1.21
	FIN		28.4 ± 3.39		27.6 ± 3.42		20 ± 2.51

Abbreviations: ACL: Angus-by-Charolais or Angus-by-Limousin crossbred animals; FIN: finishing period; GR: growth period; GR to FIN: growth to finishing interval. WEAN: after weaning; WEAN to GR: weaning to growth interval. Ages are expressed as mean ± standard deviation.

**Table 5 animals-12-01671-t005:** Average daily weight gain (kg/d) of fattening steers during three intervals.

Interval	Wagyu	Wangus	ACL	*p*-Value
WEAN to GR	0.916 (0.806–1.025) ^a^	1.046 (0.915–1.196) ^b^	1.293 (1.107–1.406) ^c^	<0.001
GR to FIN	0.628 (0.583–0.753) ^a^	0.64 (0.587–0.732) ^a^	0.802 (0.704–1.048) ^b^	0.001
WEAN to FIN	0.78 (0.73–0.85) ^a^	0.86 (0.76–0.89) ^a^	1.12 (0.92–1.20) ^b^	<0.001

Abbreviations: ACL: crossbred animals Angus-by-Charolais or Angus-by-Limousin; FIN: finishing period; GR: growing period; GR to FIN: growing to finishing interval; WEAN: after weaning; WEAN to GR: weaning to growing interval. The values are expressed as median values (interquartile range). Different superscripts a, b, and c in values of a same row indicate significant differences with the *p*-value of the last column.

**Table 6 animals-12-01671-t006:** Transversal approach of metabolic parameters of steers at four time points during the fattening period and before slaughter.

		WEAN		GR		FIN		SLAUGHTER	
		Median (IQR)	*p*-Value	Median (IQR)	*p*-Value	Median (IQR)	*p*-Value	Median (IQR)	*p*-Value
	WY	0.3 (0.2–0.4) ^a^	0.001	0.5 (0.4–0.5)	0.13	0.4 (0.4–0.5) ^a^	0.001	0.4 (0.4–0.6)	0.54
BHB (mmol/L)	WN	0.3 (0.2–0.4) ^a^		0.5 (0.4–0.5)		0.5 (0.4–0.5) ^a^		0.4 (0.3–0.5)	
	ACL	0.5 (0.4–0.5) ^b^		0.4 (0.4–0.5)		0.4 (0.3–0.4) ^b^			
	WY	0.2 (0.1–0.3)	0.2	0.2 (0.1–0.3)	0.066	0.3 (0.2–0.3) ^a^	0.019	0.2 (0.1–0.3)	0.32
NEFA (mmol/L)	WN	0.2 (0.1–0.3)		0.2 (0.2–0.4)		0.20 (0.1–0.3) ^b^		0.2 (0.2–0.4)	
	ACL	0.1 (0.1–0.4)		0.2 (0.1–0.3)		0.3 (0.2–0.3) ^a^			
	WY	110 (101–120) ^a^	0.002	95 (89.0–100)	0.84	84.5 (80.5–92.0) ^a^	0.023	90.5 (83.0–111)	0.1
GLU (mg/dL)	WN	110 (99–116) ^a^		95 (90.0–100)		87.0 (78.3–93.0) ^a^		83.0 (75.7–97.0)	
	ACL	118 (115–127) ^b^		94.1 (90.0–99.0)		93.0 (84.0–98.0) ^b^			
	WY	342 (303–389)	0.1	297 (273–318)	0.14	327 (293–362) ^a^	0.005	302 (282–364)	0.4
FRU (mg/dL)	WN	333 (297–368)		299 (274–327)		299 (269–324) ^b^		292 (272–315)	
	ACL	317 (280–345)		307 (286–333)		299 (278–326) ^b^			
	WY	24.0 (18.0–33.0) ^a^	0.001	23.0 (16.5–32.0) ^ab^	0.012	19.5 (13.0–30.5)	0.109	27.5 (13.0–41.0)	0.69
LAC (mg/dL)	WN	22.0 (17.0–37.0) ^a^		21.71 (16.0–29.5) ^a^		24.0 (17.0–37.0)		19.0 (12.5–32.3)	
	ACL	58.0 (24.8–67.0) ^b^		30.0 (20.0–39.0) ^b^		21.0 (13.0–33.0)			
	WY	139 (88.0–177) ^a^	0.027	109 (95.0–125) ^a^	0.001	139 (117–162) ^a^	0.003	148 (111–181)	0.98
TC (mg/dL)	WN	140 (113–155) ^a^		122 (106–140) ^b^		143 (118–175) ^a^		137 (124–165)	
	ACL	93.0 (80.5–117) ^b^		107 (89.0–143) ^ab^		114 (97.0–138) ^b^			
	WY	54.0 (39.0–64.0) ^ab^	0.033	42.0 (36.5–49.0) ^a^	0.003	47.5 (41.5–60.5) ^a^	0.016	60.0 (48.5–65.0)	0.97
HDL (mg/dL)	WN	59.0 (47.0–66.0) ^a^		53.47 (45.0–62.0) ^b^		56.0 (45.0–73.2) ^b^		54.0 (46.5–87.0)	
	ACL	42.0 (38.5–49.0) ^b^		43.5 (33.0–58.0) ^a^		42.0 (38.0–51.0) ^a^			
	WY	20.0 (11.0–28.0)	0.17	10.5 (7.0–14.5) ^a^	<0.001	11.0 (6.0–17.5) ^a^	0.001	8.5 (6.0–18.0)	0.04
LDL (mg/dL)	WN	22.0 (15.0–27.0)		14.0 (9.7–18.0) ^b^		18.0 (12.0–22.5) ^b^		19.7 (14.2–22.2)	
	ACL	18.0 (10.0–22.0)		15.0 (12.0–18.0) ^b^		19.0 (13.0–25.0) ^b^			
	WY	24.0 (17.0–35.0)	0.14	20.7 (16.0–29.0) ^a^	0.001	30.0 (21.8–45.0) ^ab^	0.023	29.0 (23.0–38.0)	0.23
TG (mg/dL)	WN	28.0 (20.0–39.7)		25.5 (20.0–34.5) ^b^		31.0 (25.0–45.0) ^a^		30.0 (23.5–38.5)	
	ACL	24.0 (17.0–27.0)		22.0 (15.0–28.0) ^a^		21.0 (16.0–43.0) ^b^			
	WY	20.0 (16.0–24.0) ^a^	0.037	25.0 (21.0–31.0) ^a^	0.003	24.0 (20.5–28.5) ^a^	0.001	23.0 (19.0–28.0)	0.05
Urea (mg/dL)	WN	23.0 (17.5–27.0) ^b^		29.5 (24.0–35.3) ^b^		29.0 (25.0–35.0) ^b^		28.7 (23.0–39.0)	
	ACL	20.0 (14.0–23.0) ^ab^		22.0 (20.0–25.3) ^c^		24.0 (20.0–28.0) ^a^			

Abbreviations: ACL: crossbred animals Angus-by-Charolais or Angus-by-Limousin; BHB: β-hydroxybutyrate; FIN: finishing period; FRU: fructosamine; GLU: glucose; GR: growing period; HDL: high density lipoprotein cholesterol; LAC: lactate; LDL: low density lipoprotein cholesterol; NEFA: non-esterified fatty acid; TC: serum total cholesterol; TG: triglycerides; WEAN: after weaning; WN: Wangus animals (Wagyu-by-Angus); WY: fullblood Wagyu animals. Parameters are expressed as median (interquartile range, IQR). Different superscripts a, b, and c in the values indicate significant differences among the breed groups.

## Data Availability

Not applicable.

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
