# Peer review of "Comparison of Pure and Crossbred Japanese Black Steers in Growth Performance and Metabolic Features from Birth to Slaughter at a Spanish Fattening Farm"

_animals, 2022, doi:10.3390/ani12131671_

Round 1
Reviewer 1 Report
Line 152-153, "some of the same ... and second subsets?" Confusing.
Lines 156-158, this parts need to be throughly rewritten. Please make clear what kind of TMR for each period.
Line 180, what is WEAN and GR? Also what is FIN?
Figure 1, please add SEM for the mean of bodyweight and height. The same for other figures.
Most important, could the author provide pictures of marbling for these different breeds?
Author Response
We thank the through revision that he/she did of our manuscript. We included every suggestion he/she pointed out throughout the manuscript and the changes are highlighted in yellow.

Reviewer 2 Report
The manuscript is very well written with a excellent introduction and discussion parts. The manuscript presents some new information but has limited experimental design.
The major issue is that the study, for being an longitudinal, has some limitations, such number of animals per groups, use of sample of animals in some analysis, wide range of time interval to measure body fattening rate, very limited animals evaluated in the slaughtering phase and limited number of animals in the FIN phase for ACL breed.
Also, they consider ACL as a cross breed with two different options of breed. If they are comparing Wagyu and their cross, a very specific genetic pattern, the other groups should be specific. I imagine that they did this is increase the number of animals in the ACL group, but in my opinion this weakening the manuscript, because all the statistical analysis used this limited sampling points and this impaired in the statistical results.
In my opinion the statistical analysis should consider only Wagyu and Wangus, the ACL data should be removed since is reduced and include 2 different breeds possible (Charolais and Limousin).
The authors can add the data, for comparison in the discussion, but the results sections should be separate with data with robust information and data without robust information.
Another points?
Wagyu purebreds being probably more profitable than Wangus crossbreds.
This is weird, probably? Do you evaluate this? Your conclusion must be supported by your data. In your discussion, you can comment about why you think Wagyu better.
It seems odd to me the absence of ethical approval. An experiment such this, with ultrasound and blood sampling for sure will require a approval in Brazil. But I am considering the argument presented by authors that, the know the requirements of their county.
In addition, when select a samples of the animals to be analyzed, it’s necessary to include more details, which was the criteria, how many animals were used for which analysis.
I miss a photo of the animals production system, I am pretty sure that people would be interested in see photos of those animals in this system. You can add in the supplementary material. Also, a photo make your paper when published more discoverable.
Author Response
We thank the through revision that he/she did of our manuscript. We included every suggestion he/she pointed out throughout the manuscript, and they were highlighted in green.

Reviewer 3 Report
The aim of the research was to comparison of pure and crossbred Japanese Black steers in growth performance and metabolic features from birth to slaughter age at a Spanish fattening farm. The number steers used in the experiment is sufficient. The applied research methods are correct. The discussion is well conducted and comprehensive. Well-chosen references. Before publishing in Animals, the article requires additions and corrections. The proposed changes are listed below:
General comments:
Please prepare the article in accordance with the instructions for authors.
· Please complete "Citation" on the front page
· When specifying references without spaces after consecutive reference numbers, for example [2,3] instaed of [2, 3]
· Table 1, Table 2, Table 3… .. Table 6 and Figure 1, Figure 2… Figure 5 must be in bold
· The headings in tables 1-6 must be in bold
· There must be "dot" after every parts abbrevaited name journal, for example, "Livest. Sci." instead of Livest Sci
· The article numbers must be as in any other articles in Animals
Detailed comments
L50 "usually higher" instead of higher
L50 b - hydroxybutyrate too?
L58-59 Does the IMF content affect the color and taste of beef meat?
L163 + no explanation for "Zn"
L178 with what accuracy was the BW determined
L184 „18 cm”, space after "18"
L185 „3.5 Mhz”, space after "3.5"
L235 add a „dot” at the end of the title Table 3
L243 no abbreviation "SL" in table 4, please delete the explanation
L266 ACL instead of CA
Table 5 header "Wagyu" instead of "Waygu"
Table header 5, p-Value, value capitalized
L279 abbreviation "SL" is missing in table 5, please delete the explanation
L286 GR to FIN close to 3 groups
L286 no WEAN to FIN period in Figure 2
In table 6, I propose to provide data from 1 to 100 to 1 decimal place (for example 20.0; 20.5), above the value above 100 to 1 unit (for example 101)
Table header 6, p-Value, „Value” capitalized
L336 TC for GR period also?
L337 NEFA for the GR period too?
L337 FRU for the period GR also?
L339 „in LDL and urea content (Table 6)” instead of current form
L355-357 Description inconsistent with data in Figure 4; NEFA increased in WN group?, NEFA decreased in WY steers?, BHB and LAC decereased in WN group?
L355 FRU instead of FRUC
L371 for Urea content none influence breed, but on Figure 5J a < 0.001 for breed, please correct the description
Author Response
We thank the through revision that he/she did of our manuscript. We included every suggestion he/she pointed out throughout the manuscript, and they were highlighted in blue.

Round 2
Reviewer 2 Report
The authors successfully responded to all my comments and revised the manuscript accordingly.
The justification gave by authors regarding the breed is adequate and I understand the desire to maintain the results. The authors acknowledge the limitation of the research and included a new figure as request.
In my opinion the manuscript can be accepted for publication.